# Dynamic Greenland ice sheet driven by pCO$_2$ variations across the Pliocene Pleistocene transition

Ning Tan[1,2], Jean-Baptiste Ladant[1,3,4], Gilles Ramstein [1], Christophe Dumas[1], Paul Bachem[5] & Eystein Jansen[6]

It is generally considered that the perennial glaciation of Greenland lasting several orbital cycles began around 2.7 Ma along with the intensification of Northern Hemisphere glaciation (NHG). Both data and model studies have demonstrated that a decline in atmospheric pCO$_2$ was instrumental in establishing a perennial Greenland ice sheet (GrIS), yet models have generally used simplistic pCO$_2$ constraints rather than data-inferred pCO$_2$ evolution. Here, using a method designed for the long-term coupling of climate and cryosphere models and pCO$_2$ scenarios from different studies, we highlight the pivotal role of pCO$_2$ on the GrIS expansion across the Plio-Pleistocene Transition (PPT, 3.0–2.5 Ma), in particular in the range between 280 and 320 ppm. Good qualitative agreement is obtained between various IRD reconstructions and some of the possible evolutions of the GrIS simulated by our model. Our results underline the dynamism of the GrIS waxing and waning under pCO$_2$ levels similar to or lower than today, which supports recent evidence of a dynamic GrIS during the Plio-Pleistocene.

---

[1] Laboratoire des Sciences du Climat et de l'Environnement, LSCE/IPSL, CEA-CNRS-UVSQ, Université Paris-Saclay, 91191 Gif-sur-Yvette, France. [2] Key Laboratory of Cenozoic Geology and Environment, Institute of Geology and Geophysics, Chinese Academy of Sciences, 100029 Beijing, China. [3] Département de Géosciences, École Normale Supérieure, Paris 75005, France. [4] LMD/IPSL, CNRS/ENS/UPMC, Ecole Polytechnique, Paris 75005, France. [5] Uni Research Climate, Bjerknes Centre for Climate Research, Jahnebakken 5, 5007 Bergen, Norway. [6] Dep. of Earth Science, University of Bergen, Bjerknes Center for Climate Research, Jahnebakken 5, 5007 Bergen, Norway. Correspondence and requests for materials should be addressed to N.T. (email: ning.tan@mail.iggcas.ac.cn)

The long-term trend that led to the initiation of the cyclic Northern Hemisphere glaciations (NHG) can be dated from 3.6 Ma[1]. A set of reconstructed SST records[2–4] spread across the globe describes a gradual cooling trend for the middle to late Pliocene and early Pleistocene (3.6–2.2 Ma), associated with a progressive increase in high $\delta^{18}O$ peaks of benthic foraminifera[5], reflecting both lower deep-water temperatures and increased ice volume. Terrestrial data from Lake El'gygytgyn are also consistent with long-term ocean cooling[6]. During this period, the first marked peak of IRD was found around 3.4–3.3 Ma (MIS MG2 and MIS M2) East of Greenland (ODP site 907[7]) and is interpreted as evidence of an important glacial event occurring before the establishment of the cyclic NHG around 2.7 Ma[7,8]. Significant amounts of IRD were recovered in different ocean drilling sites in the North Atlantic during the PPT[9]. These deposited IRD likely originated from Greenland, Iceland, North America and Scandinavia, suggesting large land ice expansions to the coast during this interval[7,9]. In addition, records based on volcanic and sedimentary facies from Iceland also indicate the onset of a large glaciation after 2.6 Ma[10,11]. The onset of a perennial GrIS and of the cyclic NHG at the end of the PPT stands out as a tipping point in the climate evolution of the Earth. Indeed, it marks the beginning of a low $pCO_2$ world with perennial ice sheets in both hemispheres, an infrequent occurrence in the Earth's history[12], thereby creating specific geologic and climatic conditions allowing the development of glacial/interglacial cycles.

Greenland, however, may have experienced waxing and waning of ice before the intensification of NHG, as suggested by Eocene, late Miocene and early Pliocene IRD records[13–15]. In particular, the last large Northern Hemisphere glaciation prior to the major intensification at 2.7 Ma occurred during MIS-M2 (3.312–3.264 Ma)[1,16–18]. This 50 kyrs glaciation was followed by the well-established Mid-Pliocene Warm Period (MPWP) from 3.3 Ma to 3.0 Ma for which numerous observational and modelling studies performed in the framework of the PlioMIP project[19] have demonstrated that the warmer conditions led to a reduced GrIS[20–22] (Supplementary Fig. 1). The subsequent build-up of a perennial GrIS across the PPT remains poorly constrained from a spatio-temporal point of view due to its minor contribution to the signal in global benthic foraminiferal $\delta^{18}O$ records and the paucity of direct geological data associated with the GrIS expansion. Models have the potential to provide valuable insights into the GrIS evolution but its transient nature remains complicated to explore with fully coupled Global Climate Models (GCM). Most climate/cryosphere studies on the GrIS onset during the PPT using GCM have generally produced snapshot climatic simulations subsequently used to force an ice sheet model (e.g. ref. [23]). For instance, Lunt et al.[23] demonstrated, based on a series of fully coupled GCM snapshot experiments with different forcing factors, that $pCO_2$ decline was the major driver of the GrIS glaciation. However, this result was obtained from equilibrium simulations at 2.7 Ma and the underlying GCM simulations included a pre-existent Pliocene GrIS[21]. A more recent study, also using snapshot simulations at 2.7 Ma, demonstrated that $pCO_2$ values had to remain low to counterbalance the increasing summer insolation in order to maintain the GrIS after the initial onset[24]. In contrast, low resolution, conceptual and intermediate complexity models have been used to perform transient long-term experiments[25–27], but these models remain simplified with respect to many processes and do not have the spatial resolution to focus specifically on Greenland. Willeit et al.[28] have recently worked on increasing the resolution but their simulations of the GrIS across the PPT still used pre-defined $pCO_2$ forcings. Therefore, although important insights into the dynamics of the cryosphere across the PPT have emerged from these studies, simulation of the GrIS evolution across PPT with a forward physically-based model driven by realistic forcings remains a major challenge. Here, we use a recent numerical interpolation method[29] (see Methods), which couples climate simulations obtained with the fully coupled IPSL-CM5A model[30] and the 15-km resolution version of the ice sheet model GRISLI[31], in order to investigate the transient evolution of the GrIS across the PPT. The major advantage of our approach is the ability to directly test the response of the climate-ice sheet system to different $pCO_2$ evolution scenarios (see Methods) in order to define plausible $pCO_2$ scenarios that led to the GrIS inception and variability. We thus apply various reconstructions of $pCO_2$ evolution across the PPT from both proxy records and inverse modelling studies (e.g. ref. [27,28,32–35]), as well as constant $pCO_2$ evolutions.

## Results

**GrIS sensitivity to constant $pCO_2$ scenarios.** We first present six ice sheet experiments with constant $pCO_2$ values ranging from 220 to 405 ppmv using realistic orbital variations[36] (Supplementary Fig. 2b). As shown in Fig. 1a, the evolution of the GrIS clearly shows that below 280 ppmv of $pCO_2$, it is possible to trigger and maintain a large perennial ice sheet over Greenland even during intervals of strong summer insolation, in particular around 2.6 Ma, whereas $pCO_2$ higher than 320 ppmv prevent the GrIS from remaining perennial across the whole PPT period. Between 240 and 360 ppmv, the sensitivity of the GrIS volume to orbital variations is considerable, illustrating the role of the complex interplay between $pCO_2$ and orbital variations in the dynamics of the GrIS, especially after 2.7 Ma when insolation becomes highly variable. Finally for $pCO_2$ levels above 320 ppm, the ice sheet extent remains limited across the whole 3–2.5 Ma interval. The results of these constant simulations are in good agreement but go one step beyond previous transient experiments at constant $pCO_2$[37] in that we demonstrate that the GrIS possesses a dynamism on orbital timescales across the PPT for only a narrow range of atmospheric $pCO_2$ concentrations.

**GrIS sensitivity to the initial boundary conditions.** Existing knowledge indicates that the GrIS largely retreated during the MPWP, but the exact configuration of the Greenland ice sheet and its topography remain highly uncertain[17]. In order to investigate the impact of different initial states of the GrIS on its subsequent evolution, we test two different initial boundary conditions: an ice-free Greenland (Supplementary Fig. 1c) and the modern GrIS (Supplementary Fig. 1a). We perform constant $pCO_2$ simulations at three different levels (220 ppmv, 320 ppmv, 405 ppmv). Our results show that for extreme $pCO_2$ values (220 or 405 ppm) the simulated ice sheet evolution does not depend upon the initial configuration of the ice sheet at 3.0 Ma (Fig. 1b). For $pCO_2$ levels closer to the modelled threshold for glaciation (320 ppmv), orbital and $pCO_2$ forcings have comparable influence on the simulated GrIS volume but these simulations demonstrate that the initial configuration of the ice sheet also controls the GrIS response. Indeed, bedrock vertical motion driven by the presence or absence of ice can alter topography for tens or hundreds of meters, which, at $pCO_2$ levels close to the threshold, may lead to differences in ice volume and extent. However, the general evolution of the GrIS volume remains similar in the scenario starting with an ice-free Greenland and in that starting from a full ice sheet. More importantly, during major deglaciation episodes driven by high summer insolation, the modelled GrIS in both cases recedes to identical configurations. As there is evidence for a much reduced, perhaps even non-existent, GrIS during interglacials of the MPWP (e.g., ref. [38,39]), we initialize the ice sheet

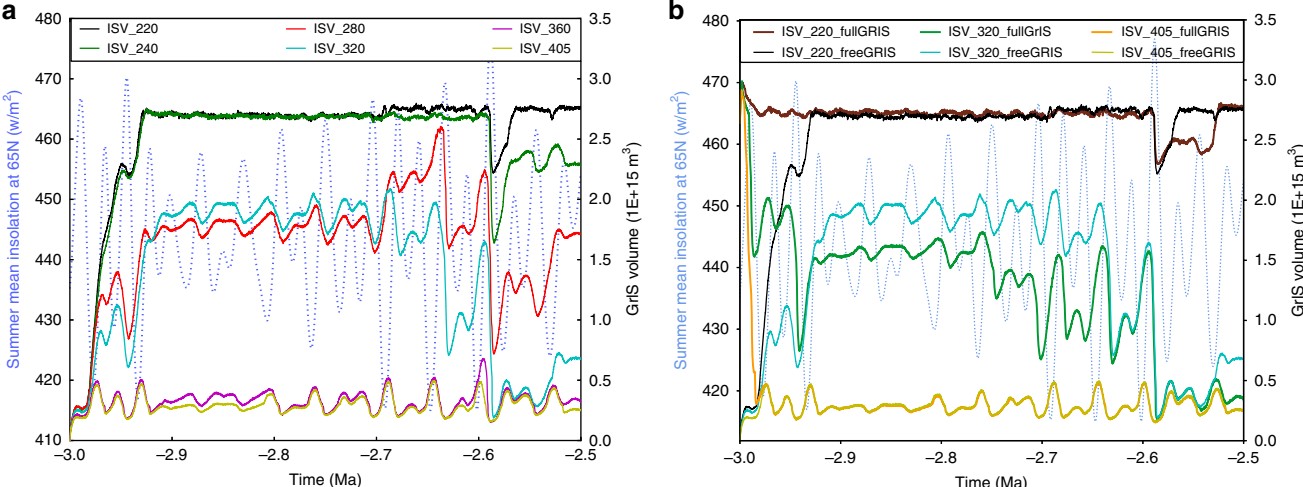

**Fig. 1** Sensitivity tests with constant pCO₂ scenarios. **a** Simulated GrIS volume (ISV on the figure) with various constant pCO₂ concentrations (220 ppmv, 240 ppmv, 280 ppmv, 320 ppmv, 360 ppmv, 405 ppmv; so that ISV_220 indicates the simulated GrIS with constant pCO₂ of 220 ppmv). **b** Simulated GrIS volume with the three pCO₂ concentrations (220 ppmv, 320 ppmv and 405 ppmv under two different initial GrIS configurations: the present-day full GrIS or an ice-free Greenland. Light blue dash lines represent the summer mean insolation at 65 °N

model with an ice-free Greenland. Indeed, the starting date of our simulation (3 Ma) is preceded by a very high summer insolation maximum (Supplementary Fig. 2b).

**GrIS evolution using pCO₂ reconstructions**. In the following, we force our set-up with published reconstructions of atmospheric pCO₂, derived from recent proxy reconstructions[32–34] or from inverse modelling[27,28,35], in order to compare the modelled GrIS evolution to previous work and to IRD records across the PPT.

We employ recent pCO₂ reconstructions based on alkenones and boron measurements[32–34] (teal lines on Fig. 2). These records depict different pCO₂ evolution scenarios from a mostly linear and low-resolution decrease for the older record[34] to high-resolution and high pCO₂ variations in the most recent record[33]. When forced by the estimated high pCO₂ levels of Seki et al.[34], the GrIS evolution confirms the major role of orbital variations when pCO₂ levels are confined to between ~280 ppmv and ~320 ppmv (Fig. 2a). While pCO₂ levels remain above 350 ppm, even large orbital variations do not significantly affect the extension of the GrIS (3.0 to 2.9 Ma). During the 2.9–2.7 Ma interval, the low variability in insolation prevents the GrIS from growing in spite of pCO₂ levels dropping to 300 ppm. The simulated GrIS evolution then shows a significant increase at 2.7 Ma and pronounced orbital scale variability between 2.7 and 2.5 Ma, but importantly, there is no full retreat of the GrIS during the insolation maximum after 2.6 Ma because this maximum is associated with pCO₂ levels below 300 ppm (Fig. 2a). In contrast, the Seki et al.[34] estimates of low pCO₂ levels generate a large perennial GrIS as early as 2.9 Ma triggered by the combination of an insolation minimum and low pCO₂. A similar modelled evolution of the GrIS is obtained when forced by the pCO₂ record of Bartoli et al.[32]. Regardless of the uncertainties on the absolute values of pCO₂ concentration, the low levels in this record force an early onset of a perennial GrIS (Fig. 2b). However, because of their low resolution, these two records do not show any pCO₂ variability on the ~10 kyrs timescale contrary to that of Martinez-Boti et al.[33], which demonstrates that the PPT pCO₂ evolution is in fact much more variable than previously thought. The uncertainties associated with the estimates of the Martinez-Boti et al.[33] record show that a completely different evolution of the GrIS can be simulated (Fig. 2c). The high (low) estimates depict

an evolution close to that driven by the high (low) estimates of Seki et al.[34], because the pCO₂ levels during times of insolation extremes (3.0–2.9 Ma and 2.7–2.6 Ma) are similar. However, the simulation forced with the mean pCO₂ estimates of Martinez-Boti et al.[33] shows an early attempt of GrIS expansion (~2.98–2.94 Ma) before a progressive onset from 2.8 Ma onwards, with a significant increase in GrIS volume around 2.72 Ma and 2.6 Ma (Fig. 3) because of the combination of low insolation and low pCO₂ levels. After 2.6 Ma, the GrIS rapidly melts down to small ice caps in the southern and southeastern margins (Figs 2c and 3) because of a simultaneous increase in summer insolation and pCO₂. The ice sheet expansion then resumes at ~2.55 Ma (A complete GrIS evolution with the mean pCO₂ estimates of Martinez-Boti et al.[33] is shown in the Supplementary Movie 1). Interestingly, the pCO₂ records of Martinez-Boti et al.[33] generate a sharp GrIS volume decrease after 2.6 Ma, regardless of the uncertainties and earlier shape of the expansion of the ice sheet because of an increase in pCO₂ values during the insolation maximum. This seems contradictory to the presence of large and perennial ice sheets from 2.7 Ma onwards but comparison with IRD records[7,9] does not invalidate a scenario with a widescale deglaciation after 2.6 Ma.

When forced by the monotonously decreasing and obliquity modulated pCO₂ scenario defined by Willeit et al.[28], our simulated GrIS shows a similar evolution to that of Willeit et al.[28] (Fig. 4), in particular in the large increases in GrIS volume reproduced at 2.7 Ma and 2.55 Ma, as well as the large decrease at 2.6 Ma, although our GrIS volume displays less variability because of a lower sensitivity to the orbital forcing in our fully coupled climate-ice sheet model. Inverse modelling studies[27,35] have also provided potential pCO₂ reconstructions for this interval (Supplementary Fig. 3). However, the low pCO₂ concentrations throughout most of these reconstructions lead to a large perennial GrIS even during the well-established MPWP warming period[19], in contrast to evidence for a much reduced GrIS[38,39].

**Discussion**

In this work, we restrict high-resolution ice sheet modelling to scenarios of the GrIS evolution across the PPT although much evidence for ice growth outside Greenland has been reported during this period[8,40,41]. This decision is warranted because Greenland is one of the primary location for ice nucleation in the

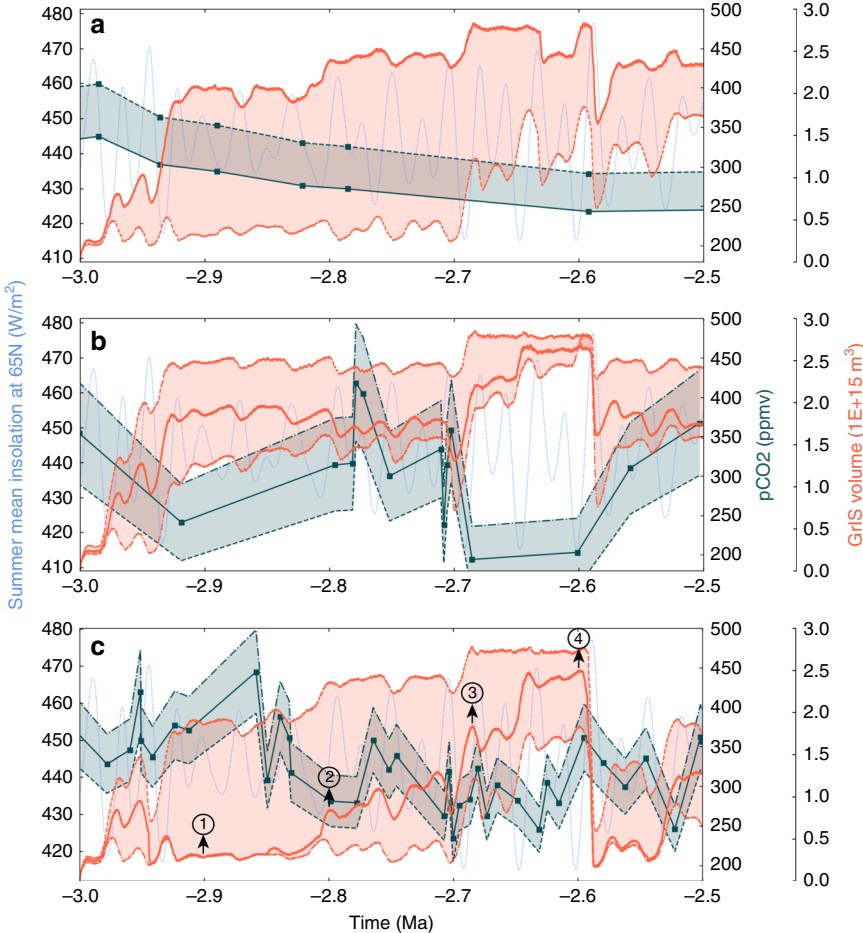

**Fig. 2** Simulated GrIS volume evolution based on different $pCO_2$ records. **a** Seki et al.[34]; **b** Bartoli et al.[32]; **c** Martinez et al.[33]. Dashed light blue line represents the boreal summer insolation at 65°N. Orange lines are the simulated GrIS volumes based on $pCO_2$ records and their uncertainties, represented by the solid teal lines and shaded teal areas. **a** Alkenone-based $pCO_2$ record of Seki et al.[34], determined using size-corrected $\varepsilon_{37:2}$ values for the modern range of *b*-values[34]. The solid (dashed) orange line is the simulated GrIS volume obtained with the solid (dashed) $pCO_2$ line. **b** Boron isotopes-based $pCO_2$ record of Bartoli et al.[32], with a 2 σ error range. The solid orange line is the simulated GrIS volume with mean $pCO_2$ values and shaded orange lines are the GrIS volumes obtained with low and high extremes of the $pCO_2$ uncertainties. **c** As in **b** but for the boron isotopes-based $pCO_2$ record of Martinez-Boti et al.[33], with an uncertainty range corresponding to 68% of 10,000 Monte Carlo simulations with full propagation of uncertainties[33]. Circled numbers at 2.9 Ma, ~2.8 Ma, ~2.72 Ma and 2.6 Ma correspond to the ice sheet snapshot displayed in Fig. 3

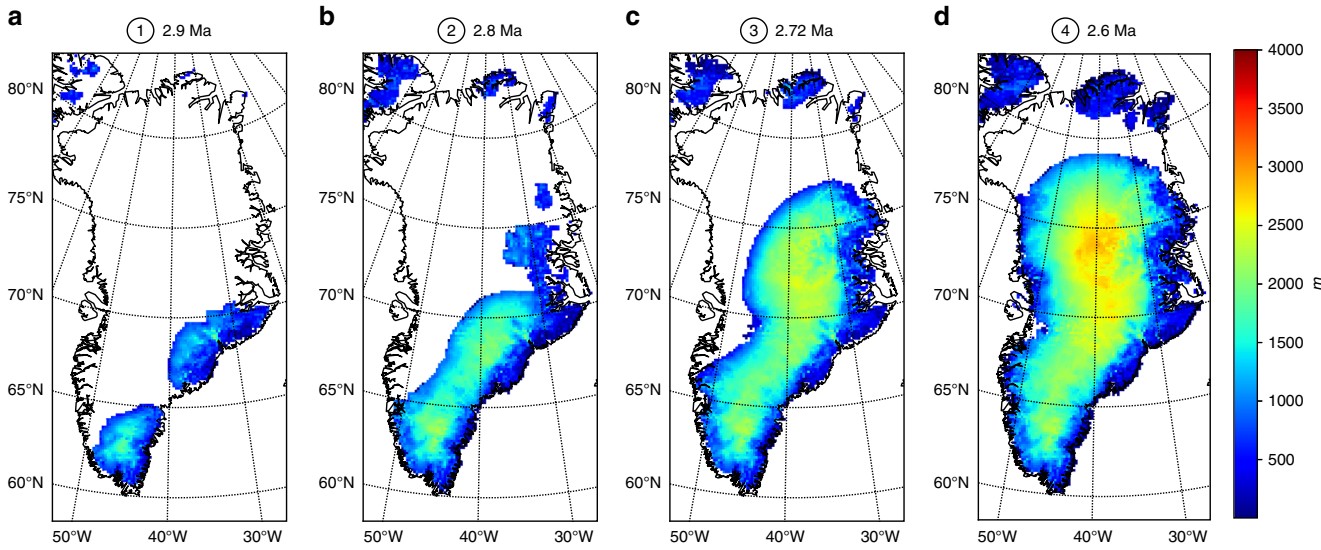

**Fig. 3** Greenland ice sheet thickness snapshots in the modelled GrIS evolution. These snapshots are taken from the simulated GrIS evolution based on the $pCO_2$ record of Martinez-Boti et al.[33] (mean) at 2.9 Ma (**a**), 2.8 Ma (**b**), 2.72 Ma (**c**) and 2.6 Ma (**d**), as shown on Fig. 2c

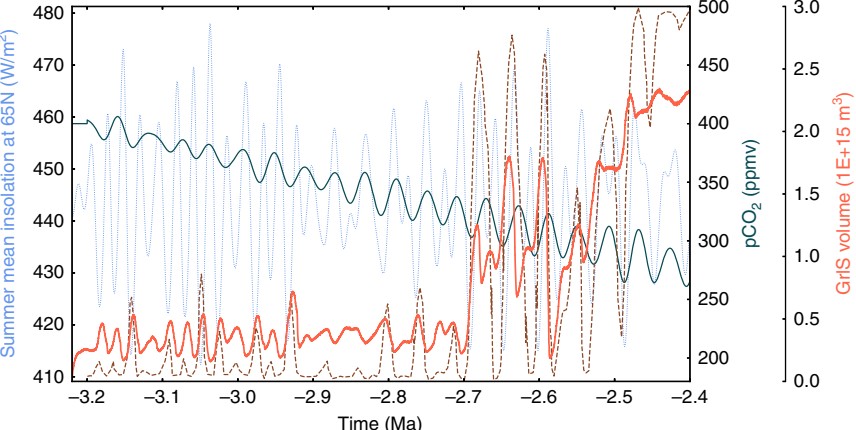

**Fig. 4** Simulated GrIS volume evolution with the pre-defined best scenario of pCO2 from Willeit et al.[28]. The brown dash line represents the GrIS evolution of Willeit et al.[28] under the same pCO2 scenario. The dashed light blue line represents the boreal summer insolation at 65°N

North Hemisphere[15] and because there is currently no agreement on the possible extent and configuration of possible ice sheets outside Greenland. However, this restriction hampers comparisons with δ18O and sea level records because the maximum ice volume that can be accommodated over Greenland represents roughly 7 m of equivalent sea level, i.e. in the range of experimental and/or instrumental errors. Significant insights can however be gained from adjacent IRD records[9] in order to further constrain the possible evolution of the GrIS and of pCO2 across the PPT. It should be kept in mind that attempting to constrain the GrIS geometry and volume from IRD records remains speculative because the absence of IRDs does not necessarily correlate with the absence of GrIS and because IRD peaks may contain material derived from other sources than the melting Greenland icebergs. In addition, iceberg trajectories may change and the ambient temperature along the iceberg trajectory may influence the melt-out rates of the IRD contained in the icebergs. For instance, variations in North Atlantic SST and/or currents or changes in the sediment contents of the calved icebergs could explain the absence of IRD even if the GrIS maintains a significant iceberg discharge[41,42]. In addition, IRD peaks may represent an ice growth phase, during which increasing GrIS volume could lead to increased iceberg discharge, as well as an ice melt phase, during which the warmer climatic conditions could lead to enhanced ice sheet melting and, consequently, enhanced ice flux at the margin. Despite these uncertainties, IRD records offer a first-order insight into ice sheet dynamics and their relationship to orbital variations, which may aid in defining the most probable scenario for GrIS evolution. We used ODP Site 907[7], IODP Site U1307[43] and ODP Site 611[41] (Fig. 5. More details about these data can be found in Supplementary Note 1) because the IRD records from these sites can be confidently assumed to originate primarily from the GrIS[41]. Sites U1307 and 907 are located offshore from Greenland margins and show small but continuous IRD deposition from as early as 3 Ma (Fig. 5b), with the exception of a single peak at 2.92 Ma at Site 907. Around 2.7 Ma, several IRD peaks in both records suggest intensification of the iceberg discharge, broadly correlated to North Atlantic sea surface temperatures (SST) cooling events (Fig. 5c). Accordingly, the record from Site 611, located further south from Greenland, shows an absence of IRD deposition before 2.72 Ma and several peaks afterwards.

There is variable agreement between our reconstructed GrIS volume scenarios and inferred GrIS extension as implied by IRD records. For instance, the GrIS evolution forced by the pCO2 record of Bartoli et al.[32] agrees poorly with the IRD records

because it suggests a large to near-complete perennial GrIS as early as 2.98–2.95 Ma (Fig. 2b). Around 2.9 Ma, a large ice sheet covering the totality of the southern Greenland margin is present in the results forced by the pCO2 records of Bartoli et al.[32]. However, there are not any particular changes above the 3.0–2.7 Ma background IRD values at Site U1307. At Site 907, an early increase of a large GrIS could explain the 2.92 IRD peak, but in this case the IRD deposition should have continued with higher levels relative to the pre-glaciation interval. In addition, the absence of any IRD at Site 611 at that time suggests that the GrIS was still of relatively limited size. In contrast to the pCO2 record of Bartoli et al.[32], the mean and high pCO2 estimates of Martinez-Boti et al.[33] and the high estimate of Seki et al.[34] generate GrIS evolutions more consistent with the IRD records. The modelled GrIS using these reconstructions is limited to small ice caps on the southern and southeastern margins of Greenland during the interval 3.0–2.7 Ma interval, which agrees well with small but continuous IRD inputs at Site 907 and U1307 (Fig. 2a, c, Fig. 3 and Fig. 5a, b). The sharp increase in IRD deposition at 2.7 Ma is also accounted for by these GrIS evolution scenarios. Furthermore, the IRD records of Site 611[41] displays four IRD peaks (at ~2.7 Ma, ~2.64 Ma, ~2.6 Ma and ~2.52 Ma) that are relatively well correlated in time with decreases in North Atlantic SSTs and large modelled GrIS variations (Fig. 5b, c), in particular for the Martinez-Boti et al.[33] mean and high pCO2 scenario. Importantly, the large GrIS volume decrease after 2.6 Ma in these scenarios is not at odds with the IRD records, in that the minimal GrIS state still reaches the southern and southeastern margins of Greenland, allowing small but uninterrupted iceberg discharge at Site 907. These three pCO2 scenarios (Martinez et al.[33] high, mean and Seki et al.[34] high pCO2 estimates) therefore seem to correlate relatively well with IRD records because the GrIS expands preferentially from the south and east regions and intervals of large GrIS variations are synchronous with IRD peaks.

There is evidence however showing that after 2.7 Ma, ice sheets were not restricted to Greenland (e.g., ref. [40,41]). Although by model design ice growth in other regions of the Northern Hemisphere is not simulated, simple inferences of the impact of North American and Scandinavian ice sheets after 2.7 Ma would tend to suggest that the large GrIS retreat around 2.6 Ma might have been more limited. Indeed, the cooling effects of such ice masses on regional and global climate would presumably have partly counterbalanced the combined increase in pCO2 and summer insolation. Still, the evidence for other NH ice sheets can tentatively be used to suggest that the GrIS evolution driven by the Martinez-Boti et al.[33] mean pCO2 estimates is the most

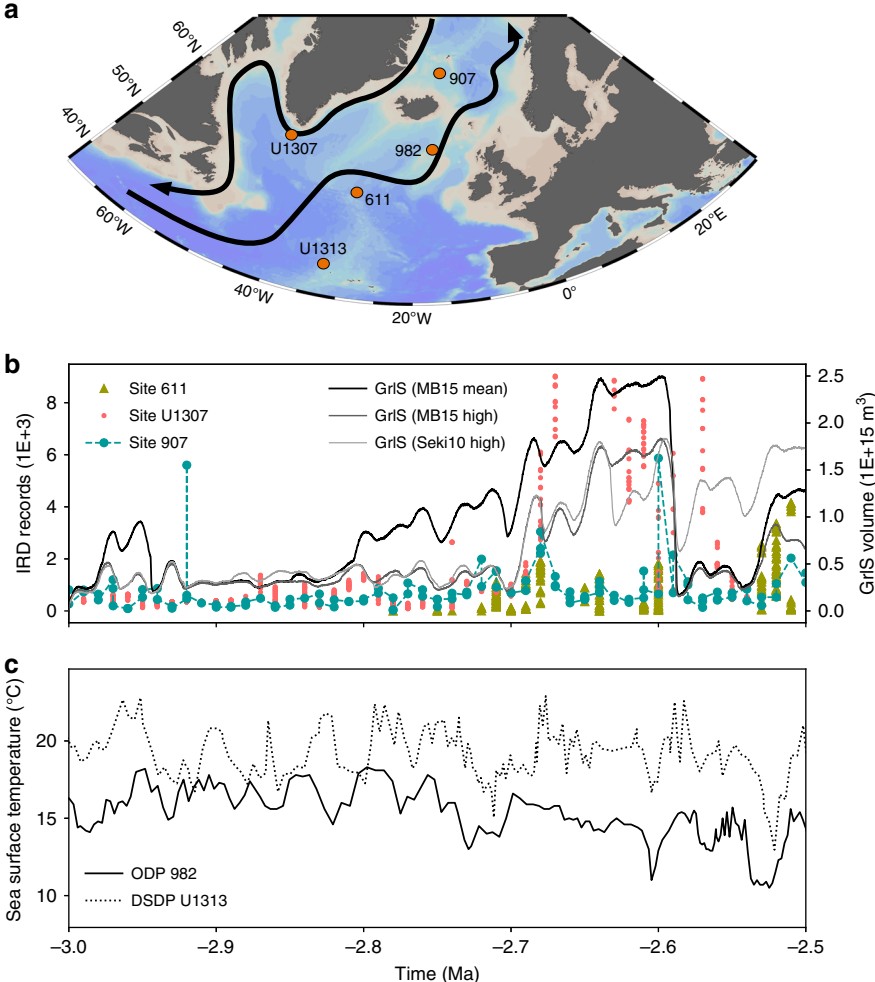

**Fig. 5** The comparison between the simulated GrIS and adjacent IRD records. **a** Map showing simplified main ocean currents in NATL and the ocean deep drilling sites referenced in this study (Ocean Data View, Schlitzer, R., Ocean Data View, odv.awi.de, 2018). **b** Related IRD records in the North Atlantic regions: site 907[7]; site U1307[43]; site 611[41] and simulated GrIS volume with the mean and high pCO₂ estimates of Martinez-Boti et al.[33] and the high pCO₂ estimate of Seki et al.[34]; **c** Reconstructed sea surface temperature at ODP site 982 by Lawrence et al.[2] and IODP site U1313 by Naafs et al.[63]

plausible pCO₂ scenario among those that are the most consistent with IRD records (i.e. Martinez-Boti et al.[33] mean and high, Seki et al.[34] high estimates), because it leads to a much larger GrIS between 2.8 and 2.6 Ma (Fig. 2c). The presence of ice sheets over North America and Scandinavia after 2.7 Ma is thus likely to be more consistent with a largely or fully glaciated Greenland because ice preferentially nucleates on Greenland before potentially expanding over other regions of the Northern Hemisphere.

In conclusion, modelling the GrIS evolution across the PPT using a recent method specifically designed for transient ice sheet experiments shows that the long-lasting paradigm explaining the large GrIS onset occurring at 2.7 Ma due to a minimum of summer insolation at 65°N is, by far, too simplistic. In our simulations, the GrIS volume appears very sensitive to pCO₂ changes during the PPT interval. Our experiments demonstrate that pCO₂ levels have to remain below 320 ppmv in order to develop and maintain perennial large GrIS across the whole PPT interval, although the absolute pCO₂ values may be model dependent. When forced by the most recent pCO₂ reconstruction available[33] for this interval, the simulated GrIS shows a good agreement with IRD records from adjacent locations[7,41,43]. The evolution of the GrIS across the PPT under this pCO₂ scenario shows large variations of ice volume of continent-sized amplitude

but indicates a continuous presence of ice even during intervals of higher pCO₂ and stronger summer insolation. This result supports recent work suggesting a very dynamic, yet persistent GrIS for the last millions of years[44], possibly receding to very small ice centres[45]. Ultimately, our work emphasizes the crucial role of pCO₂ in shaping the evolution of the cryosphere across the PPT and provides numerical arguments for a vigorously dynamic GrIS under pCO₂ levels similar to, or lower than modern values.

## Methods

**Model description**. The climate model used in this study is the IPSL-CM5A GCM[30]. The atmosphere component is the LMDZ5A version of the LMDz model (including the ORCHIDEE land-surface model) with a resolution of 3.75°×1.875° and 39 vertical layers. More details about the physical parameterization can be found in[30,46]. The ocean model is NEMOv3.2[47], which integrates the dynamical model OPA, the LIM2 sea ice model and the PISCES biogeochemical model. NEMO runs on a tri-polar grid (one pole in the Southern Hemisphere under Antarctica and two poles in the Northern Hemisphere under North America and Asia) in order to improve the representation of ocean dynamics in the northern high-latitudes. There are 31 unequally spaced vertical levels and a nominal resolution of 2° that is refined up to 0.5° in the equatorial area. The atmosphere and ocean models are linked through the coupler OASIS[48], ensuring energy and water conservation. Additional details about the IPSL-CM5A model can be found in Dufresne et al.[30].

The ice sheet model used in this study is the GRenoble Ice-Shelf and Land-Ice model (GRISLI). GRISLI is a three-dimensional thermo-mechanical model that

simulates the evolution of ice sheet geometry (extension and thickness) and the coupled temperature–velocity fields in response to climate forcing. A comprehensive description of the model can be found in ref. [31] and ref. [49]. In this study, the GRISLI model is nested to the Greenland region on a cartesian grid of 15 km × 15 km. Over the grounded part of the ice sheet, the ice flow resulting from internal deformation is governed by the shallow-ice approximation[50]. The model also deals with ice flow through ice shelves using the shallow-shelf approximation[51] and predict the large-scale characteristics of ice streams using criteria based on the effective pressure and hydraulic load. At each time step, the velocity and vertical profiles of temperature in the ice are computed, as well as the new geometry of the ice sheet. The isostatic adjustment of the bedrock in response to the ice load is governed by the flow of the asthenosphere, with a characteristic time constant of 3000 years, and by the rigidity of the lithosphere. The temperature field is computed both in the ice and in the bedrock by solving a time-dependent heat equation. Here, as the ice sheet model GRISLI is not synchronously coupled with the IPSL-CM5A model, temperature and precipitation fields are asynchronously passed from IPSL-CM5A to GRISLI. The surface mass balance is defined as the sum between accumulation and ablation computed by the positive degree-day (PDD) method[52].

**Interpolation method**. In the absence of synchronous coupling between GRISLI and IPSL-CM5A, we utilise an interpolation that asynchronously couples both models and offers the possibility of carrying long-term numerical integration of the ice sheet model while accounting for the time evolution of the main climate forcings. This method consists of building a matrix of possible climate states that are generated by IPSL-CM5A under various combinations of forcings, which are chosen from the range of possible values taken by forcings. The ice sheet model GRISLI can then be continuously forced by temperature and precipitation fields obtained by interpolation between the different IPSL-CM5A climatic states, interpolation based on the time evolution of the forcings. The principle of this method has been described in details in Pollard (2010) and an early version of it has been applied to the Eocene-Oligocene Transition (EOT) in Antarctica[53]. In this work, we use an improved version of this method, which has first been applied to the EO glaciation[29] and that we have adapted to Greenland. Specifically here, we build a three-dimensional matrix to account for the three main drivers of an ice sheet evolution that are: (1) realistic insolation variations, (2) the atmospheric $pCO_2$ evolution and (3) the ice sheet feedbacks on itself. The matrix hence comprises temperature and precipitation fields that are obtained from reference IPSL-CM5A runs initialized with different combinations of orbital parameters, $pCO_2$ and ice sheet size. In a second step, the temperature and precipitation fields that force GRISLI can be computed by interpolating between the reference IPSL-CM5A climatic states, based on the current value taken by the 65°N summer insolation and by the atmospheric $pCO_2$ and on the instantaneous size of the ice sheet in GRISLI. This ensures that the climatic fields passed to GRISLI are appropriately updated at each time step to follow the evolution of both external (insolation and $pCO_2$) and internal (ice sheet geometry) forcings.

**Experiment design**. In this study, we have chosen two orbital configurations that produce respectively the maximal (warm orbit) and the minimal (cold orbit) mean summer insolation at 65°N between 3.0 and 2.5 Ma, according to the astronomical solution calculated by the model of ref. [36]. At each time step of the ice sheet model simulations for the 3–2.5 Ma period, the impact of the summer insolation can be included by appropriately interpolating between reference IPSL-CM5A runs with warm or cold orbits.

The reference IPSL-CM5A runs are initialised with four different $pCO_2$ concentrations: 220 ppmv, 280 ppmv, 360 ppmv and 405 ppmv. Late Pliocene $pCO_2$ records indeed document a range of variation of the atmospheric $pCO_2$ comprised between ~200 ppmv and ~400 ppmv (See in Supplementary Fig. 2). Similarly, the instantaneous value of $pCO_2$ over the course of the simulation for the 3–2.5 Ma periods can be interpolated between reference runs with aforementioned $pCO_2$ values. As $pCO_2$ and temperature are linked via a logarithmic relationship, we prescribe a logarithmic interpolation between fixed $pCO_2$ reference runs. Conversely, the interpolation is kept linear for the insolation.

To obtain the reference Greenland ice sheet sizes that are prescribed in the reference IPSL-CM5A runs, we carry out a preliminary experiment in which we model the ice sheet development in an offline, one-way regrowth experiment (see ref. [24] for details). We initialise IPSL-CM5A with standard PlioMIP phase 1 conditions[54], which are then modified to start from an ice-free Greenland and extremely favourable conditions for glacial inception represented by the cold orbit described above and a $pCO_2$ concentration of 220 ppmv. GRISLI is then force with constant temperature and precipitation fields from the IPSL-CM5A simulation until the ice sheet reaches equilibrium. The ice sheet geometry is then fed back to IPSL-CM5A while $pCO_2$ and orbital parameters are kept identical. New climatic fields are thus obtained and GRISLI further simulates the ice sheet gain until a new equilibrium is reached. The geometry of the ice sheet can then be passed again to IPSL-CM5A. This process is repeated until a new iteration does not markedly increase the ice volume. Here, seven iterations allow us to obtain seven Greenland ice sheet sizes ranging from very small to nearly full size ice sheet (Supplementary Fig. 1c–i).

The matrix of reference IPSL-CM5A climatic states is then built using the forcings described above. A total of 2 (for the insolation) × 4 (for $pCO_2$) × 7 (for Greenland ice sheet sizes) simulations are run in parallel to provide reference T and P fields that cover the range of possible variations of the three main drivers that are insolation, $pCO_2$ and ice sheet size. A continuous T and P forcing can then be calculated based on the prescribed (for insolation and $pCO_2$) and emerging (for ice sheet) evolutions of these drivers. Although complex and fastidious to implement, one particular advantage of this method is that it allows virtually any $pCO_2$ scenario to be tested without additional GCM runs.

It should be noted that vegetation feedbacks linked to vegetation changes under Late Pliocene conditions are taken into account in our IPSL-CM5A simulations, since the tundra-taiga feedback has been shown to play a role in the onset of NH glaciation[20]. We divide the IPSL-CM5A boundary conditions into three types relative to their presumed impact on vegetation: cold, intermediate and warm. The cold conditions are defined by the combination of the cold orbit and either 220 ppmv or 280 ppmv of $pCO_2$ concentration. In the reference IPSL-CM5A simulations whose boundary conditions fall under the cold criterion, we modify the PlioMIP vegetation map[55] by specifying tundra north of 50°N. The intermediate conditions are defined by the combination of the cold orbit and either 360 ppmv or 405 ppmv of $pCO_2$ concentration and the PlioMIP vegetation map is modified by specifying tundra north of 65°N. Finally, the warm conditions are defined by the warm orbit, regardless of the $pCO_2$ concentration. Under these conditions, we keep the PlioMIP vegetation map unchanged. All the reference AOGCM experiments are summarized in Supplementary Table. 1.

**Model evaluation**. We assess our model performances and our modelling strategy by presenting first a comparison between the simulated surface mass balance (SMB) obtained from the CMIP5 historical IPSL-CM5A experiment[30] and averaged over the period 1981–2005 and the SMB computed with the state-of-the-art regional polar model MAR[56] forced by reanalyses over the same interval. Second, we present a sensitivity experiment of the Eemian deglaciation (150–110 Ka) in which the $pCO_2$ evolution is very well constrained.

Supplementary Fig. 4 shows that the simulated SMB with our climate model IPSL-CM5A is consistent with the reconstructed SMB from MAR, except in some regions of the northwest and southeast Greenland. These differences can be attributed: (1) to the coarser resolution of IPSL-CM5A compared to MAR, which provide some limitations in the model's ability to accurately simulate precipitation in mountainous regions (a common problem in fully coupled climate models, see e.g. ref. [57]), such as South Greenland here; 2) to biases in simulated temperatures and precipitations in the Greenland region in IPSL-CM5A[30] and/or MAR. Yet overall, the simulated SMB using IPSL-CM5A is reasonably close to that of MAR.

In order to simulate the Eemian deglaciation, we have performed four additional climate simulations with a LGM ice sheet configuration, two low $pCO_2$ scenarios (185 ppmv and 220 ppmv) and our two extreme orbital configurations. We also restrict the ice sheet simulation to Greenland. Because the climate states added in the matrix have extensive ice sheet and sea ice covers over the Northern Hemisphere, we allow the ice sheet model to use variable basal melting factors following the multi-proxy index of Quiquet et al.[58]. Basal melting factors thus vary linearly between 0 and 5 m/year in regions where depth are lower than 1300 m and are fixed to 10 m/year where depth greater than 1300 m. In addition, we use the sea level curve of Waelbroeck et al.[59] as a forcing to determine which ice model grid points are floating or grounded. These two processes are added to take into account the climatic influence of the LIG deglaciation outside Greenland because we do not explicitly model this ice sheet evolution. Finally, the ice sheet model is initialized with the LGM GrIS.

The results from this sensitivity experiment are shown on Supplementary Fig. 5. The combined increase in summer insolation and $pCO_2$ leads to the waning of the GrIS between 133 and 123 Ka. The timing of the deglaciation is consistent with results from other studies (e.g., ref. [60–62]). Stone et al.[60] "best score" simulation places the start of the GrIS deglaciation at 135 Ka (their Fig. 6b, black lines) and the end around 123 Ka. Goelzer et al.[62] restrict their figures to the interval 130–115 Ka so that the start of the deglaciation in their simulations is unclear but the lowest GrIS volume is also found around 123 Ka. Finally, Bradley et al.[61] also place the beginning and end of the GrIS deglaciation at 133 and 123 Ka, respectively (their Fig. 2). The contribution of the GrIS to sea level rise during the LIG is poorly constrained and estimated between 0.6 and 3.5 m[61]. Our simulated minimal GrIS reaches a volume of $2.846 \times 10^{15}$ km3 (Supplementary Fig. 6), which is slightly less than the equilibrium GrIS volume ($2.87 \times 10^{15}$ km3) simulated by GRISLI under preindustrial IPSL-CM5A forcings[30]. Compared to other studies, our simulated LIG GrIS does not reach a volume low enough. However, considering that we just adapted a method that is designed for the PPT and not specifically for the Eemian, we argue that the model does a reasonable job in reproducing the Eemian deglaciation. In summary, the reasonable agreement between our modelled deglaciation and the results from other transient LIG simulations provides confidence in our modelling strategy.

**Code availability**. The codes of the IPSLCM5 and GRISLI models are available on request to the authors.

## Data availability

Data that support the results of this study are available on request to the authors.

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

## Acknowledgements

We thank Camille Contoux for providing the initial ice-free Greenland bedrock configuration and Zhongshi Zhang, Alan Haywood and Florence Colleoni for discussions. We also thank Matteo Willeit for providing their model results, Masa Kageyama and Aurélien Quiquet for their help with the set-up of the Eemian sensitivity simulation and Mary Minnock for her help on improving the manuscript. This study was performed using HPC resources from GENCI-TGCC (Grant 2016-GENCI t2016012212) and supported by the French project LEFE "ComPreNdrE" (A$_2$016–992936), French State Program Investissements d'Avenir (managed by ANR), ANR HADOC project, grant ANR-17-CE31-0010 of the French National Research Agency and the Norwegian Project "OCCP" (NFR project number 221712).

## Author contributions

N.T. carried out the modelling experiments and prepared for the first manuscript, J.-B.L. and C.D. contributed equally to the experiment design and the analysis of the results. G.R. and J.-B.L. helped to improve the paper. P.B. and E.J. helped on the data-model comparison work.

## Additional information

**Competing interests:** The authors declare no competing interests.

