## [Peer Review File · Nature Communications]

Reviewers' comments:

Reviewer #1 (Remarks to the Author):

found this to be an interesting and well constructed paper that is entirely suitable for publication in "Nature Communication". The references might be broadened to include "older" (i.e. 1990's!) references on Iceland, which are indeed pertinent in a regional context.

The basic premise of this paper is to conduct a series of experiments to ascertain the sensitivity of the growth of the Greenland Ice Sheet to changes in the CO₂ content of the atmosphere, which is argued to be one of the main controls on the necessary conditions for the development of the ice sheet. I found that the arguments and attached figures to be relatively convincing. However, one basic background feature that I did not see discussed was the "platform" for the growth of the ice sheet. By this I mean--what was the elevation model of Greenland that they used and how does it differ from the current bedrock elevations? I would assume that they have allowed for glacial isostatic recovery and 2-3 Ma of glacial erosion? In my experiments with modeling the growth of the Laurentide Ice sheet (1976) the hypsometry of Baffin Is was a fundamental factor in how quickly we could get the ice sheet to grow.

I found the video of the ice sheet growth to be interesting, especially as the ice sheet appears to be centered on the SE sector of Greenland and that the high northern uplands show a relatively late response on the ice sheet. I find this both interesting and possibly strange in-so-far-as my understanding is that the Arctic Ocean does record a Pliocene warm interval. If this is the case then I would expect that the northern areas would get a substantial increase in snowfall. Infact I would strongly recommend that a figure be included in the main paper that shows the geography of the ice sheet growth with respect to Greenland.

Eiriksson, J., and Geirsdottir, A. 1991. A record of Pliocene and Pleistocene glaciations and climatic changes in the North Atlantic based on variations in volcanic and sedimentary facies in Iceland. *Marine Geology* 101: 147-159.

Geirsdottir, A., and Eiriksson, J. 1993. Growth of an Intermittent Ice Sheet in Iceland during the Late Pliocene and Early Pleistocene. *Quaternary Research* 42: 115-130.

Funder, S., Bennike, O., Mogensen, G.S., Noe-Nygaard, B., Schack Pedersen, S.A., and Petersen, K.S. 1984. The Kap København Formation, a late Cainozoic sedimentary sequence in North Greenland. *Rapp. Grønlands geol. Unders.* 120: 9-18.

Reviewer #2 (Remarks to the Author):

In their paper, Tan and coauthors present new model simulations of the evolution of the Greenland ice sheet during the Pliocene-Pleistocene transition (PPT) between 3.0 and 2.5 million years ago. Several proxies indicate that the initiation and intensification of the Northern Hemisphere (NH) glaciation occurred over this time period. Consequently, this time interval has received a lot of attention in the past decades. Many studies suggest that the atmospheric CO₂ concentration plays a major role in the initiation of glaciation both for the NH and for Antarctica. Since the available CO₂ reconstructions for the PPT are very uncertain, models have been used to explore the critical CO₂ threshold for Greenland and more generally NH glaciation. In this paper, the authors present transient simulations with an ice sheet model applied to Greenland and driven by snapshot climate model simulations for different boundary conditions (atmospheric CO₂, orbital configuration and Greenland ice volume). The model is then driven by different CO₂ scenarios, which are based on available reconstructions or inverse modelling results. They then use the ice rafted debris from several marine cores around Greenland to constrain plausible CO₂ pathways.

General comments

Although the study by Tan et al. presents interesting results and contributes to progress our

understanding of the PPT and in particular the glacial inception over Greenland, I have some concerns about the methods and the conclusions which are drawn in the paper:

- No evaluation of the model is shown. Showing that the model can reproduce the present surface mass balance of the Greenland ice sheet would add credibility to the presented results. In particular there are different regional climate modelling results for the present day surface mass balance that could be used for the evaluation (e.g. MAR, RACMO). Although a good model performance for the present day is not a guarantee that the model will perform well under very different conditions, it is a minimal requirement.
- No model uncertainties are considered. I understand that large uncertainties in the evolution of the Greenland ice sheet over the PPT are associated with the poorly constrained CO₂ concentrations, but possible uncertainties in the model should also be quantified. In particular, the matrix method used to derive the climate forcing applied to the ice sheet model is based on a limited number of climate model simulations and can not capture all non-linearities in the system. What would be the impact of having more/less climate states (e.g. an additional intermediate orbital configuration or less CO₂ concentrations)?
- In order to constrain the CO₂ evolution during the PPT, in the paper the authors compare the modelled Greenland ice volume with IRDs at different locations around Greenland. This comparison is very qualitative (Fig. 4b). Such a rough comparison can be used to rule out some extreme CO₂ scenarios, but nothing more. One way to improve the comparison would be to compare IRDs with calving from the ice sheet model into the ocean (if available). As also briefly discussed by the authors, the relation between IRDs and total ice volume is probably not linear. If the main aim of the paper, as mentioned in the title, is to constrain CO₂ pathways over the PPT, a figure showing all CO₂ reconstructions and their constrained range would be helpful.
- There is compelling evidence that after 2.7 Ma ice sheets were not limited to Greenland but covered also parts of Northern North America and Scandinavia during glacial times. The effect of ice covered areas outside of Greenland is not included in the model setup. The authors should at least discuss what implications that might have for the presented results.

Specific comments

Lines 148-153: reference to Fig. 3 is missing.

Line 198: reference to Fig.3a-c seems inappropriate.

Figure 2 is very hard to read. I would suggest splitting each panel into two, one showing the forcing and the other the response.

Figure 4b: why are there multiple points at each time for the IRD records?

English should be improved throughout the paper.

Reviewer #3 (Remarks to the Author):

This paper explores changes in the Greenland ice sheet through the Pliocene-Pleistocene transition. A coupled climate – ice sheet model is used to simulate the response of the ice sheet to varying orbital and CO₂ forcing. Several different CO₂ scenarios are used, corresponding to some idealised constant forcings and some scenarios from the published literature. Finally, the model-predicted ice volumes are compared with IRD records. The main conclusions that I took away are that (1) between certain CO₂ thresholds the ice sheet is very sensitive to orbital forcing, and (2) the Martinez-Boti and Seki CO₂ reconstructions give ice sheets that are most consistent with the IRD data.

Major Comments

In order to have confidence in the modelling strategy, an evaluation in the form of a simulation of a reasonably well known case study is needed, such as the last deglaciation (e.g. does a transient simulation of the last deglaciation end up with a reasonable modern ice sheet at present), or the

last interglacial (e.g. how does the model compare with the Kopp Greenland record). In particular I am slightly concerned that only two different orbits (“warm orbit” and “cold orbit”) are used in the GCM that is used to construct the forcing matrix. This may result in a major simplification of the complexities of the orbital forcing over the time period 3 – 2.5 Ma. As such, some evaluation of the method is needed. Either this should be included in the paper, or it should be noted clearly in the paper that the methodology is relatively untested for Greenland.

It is well known that ice sheet models are highly sensitive to poorly-constrained model parameters. For example, Stone et al (2010) show very different ice sheet responses to CO₂ forcing depending on model parameters, all of which give good agreement with modern ice sheets under modern forcing. As such, the CO₂ thresholds quoted here are themselves somewhat poorly constrained. Gasson et al (2014) found a huge range in CO₂ threshold for Antarctic glaciation from different ice sheet models with different parameters. As such, I think it is very important either to explore this sensitivity in the paper, or it should be noted clearly in the paper that the results (and in particular the numerical values of the thresholds) may be highly dependent on the internal ice sheet model parameters.

I think it would be very interesting to see a few longer simulations, that started earlier and finished later (e.g. from 3.5 Ma to 2 Ma). This would indicate whether the transition seen in these simulations is unique, or occurs at other time periods with similar orbital forcing. Some of the CO₂ records used would be suitable for this.

There were a lot of typos and grammatical errors. I noted these in a paper version, but the lack of line numbering in the submitted manuscript makes highlighting them all in this review very time-consuming. It also makes reviewing the manuscript harder, so please use line and page numbers in future.

Specific Comments

The title is unclear. How about “Sensitivity of the Greenland ice sheet to CO₂ at the Plio-Pleistocene transition”

The term “pathways” is odd...maybe “scenarios” or “records” when talking about the proxies.

Throughout, “favorable” should be clarified as “favourable for glacial inception”

Throughout “confronting” should be “comparing”

In the last line of the abstract, and towards the end of the conclusions, the relevance for future climate is briefly mentioned. This should be removed as it is not explored at all in the paper. Or alternatively it should be greatly expanded on in the main paper.

What is really meant by a “perennial” ice sheet should be clearly defined the first time it is used.

I don’t understand the sentence that begins “Pioneering studies were carried out...”

Summer insolation is not “perfectly” constrained, but “very well” constrained.

Careful when discussing thresholds. E.g. “levels higher than 320” is stated when really all you know is that the threshold is somewhere between 320 and 360, it may be 359 ppmv!

The initial condition testing is done for the two extremes, but the most interesting cases are the intermediate CO₂ levels – it would be more informative to see the results from e.g. 320 ppmv with initial states of full and no ice sheet.

It is odd that you discuss the third record (Seki) first – maybe change the panel ordering of Figure 2.

I don't agree that the comparison with Willeit et al "provides confidence in the ability of the model". All it shows is that you get similar results to another modelling study with a similar model.

The final paragraph before the conclusions is odd. I don't understand the sentence beginning "However, the integration" or the sentence beginning "Indeed, evidences".

In the conclusion I don't understand why you can infer from your results that CO₂ has to be in a narrow window, or really what "trigger and maintain" really means.

Figure 4: For the IRD, it is not clear if an absence of data (e.g. around 2.55 Ma to 2.6 Ma at site 611) means no IRD, or just no record.

Figure S3 caption – presents should be represents.

Figure 2 – what are the uncertainties in CO₂? Do they represent a 1-sigma, 2-sigma, or range, or what?

Stone et al (2010). Investigating the sensitivity of numerical model simulations of the modern state of the Greenland ice-sheet and its future response to climate change. *The Cryosphere*, 4, 397-417.

Gasson et al (2014). Uncertainties in the modelled CO₂ threshold for Antarctic glaciation, *Clim. Past*, 10, 451-466.

(Summaries: In this response letter, there are eight figures to illustrate our response. Figure.2 and Figure.6 are also included in our Supplementary Materials)

Reviewer #1 (Remarks to the Author):

I found this to be an interesting and well constructed paper that is entirely suitable for publication in "Nature Communication". The references might be broadened to include "older" (i.e. 1990's!) references on Iceland, which are indeed pertinent in a regional context.

Response: Thanks for these positive comments. We have added the suggested references on Iceland ice sheet in the introduction (line 47).

The basic premise of this paper is to conduct a series of experiments to ascertain the sensitivity of the growth of the Greenland Ice Sheet to changes in the CO₂ content of the atmosphere, which is argued to be one of the main controls on the necessary conditions for the development of the ice sheet. I found that the arguments and attached figures to be relatively convincing. However, one basic background feature that I did not see discussed was the "platform" for the growth of the ice sheet. By this I mean---what was the elevation model of Greenland that they used and how does it differ from the current bedrock elevations? I would assume that they have allowed for glacial isostatic recovery and 2-3 Ma of glacial erosion? In my experiments with modeling the growth of the Laurentide Ice sheet (1976) the hypsometry of Baffin Is was a fundamental factor in how quickly we could get the ice sheet to grow.

Response:

The initial bedrock map used is based on the isostatically-rebounded ice-free Greenland topography (GrIS) from the PRISM3 conditions (Dolan et al. 2012). Figure 1 shows the surface elevation and bedrock maps for an initial ice-free Greenland and for a large modeled Greenland ice sheet.

In addition, the GRISLI ice sheet model accounts for the isostatic response using the ELRA method (Elastic Lithosphere-Relaxed Asthenosphere). The isostatic adjustment of bedrock in response to ice load is governed by the flow of the asthenosphere, with a characteristic time constant of 3000 years, and by the rigidity of the lithosphere (see Ritz et al. 2001).

Figure.1 Surface elevations and underlying bedrocks for an initial ice-free Greenland and for a large GrIS .

I found the video of the ice sheet growth to be interesting, especially as the ice sheet appears to be centered on the SE sector of Greenland and that the high northern uplands show a relatively late response on the ice sheet. I find this both interesting and possibly strange in-so-far-as my understanding is that the Arctic Ocean does record a Pliocene warm interval. If this is the case then I would expect that the northern areas would get a substantial increase in snowfall. In fact I would strongly recommend that a figure be included in the main paper that shows the geography of the ice sheet growth with respect to Greenland.

Response:

Observational studies (e.g., Dowsett et al. 2013) indeed show that the Arctic Ocean was warmer during the Pliocene warm interval, which have increased precipitations in the northern areas of Greenland. However, results from the PlioMIP modeling intercomparison project show that models underestimate the mid-Pliocene high latitude warming relative to that inferred from data (Haywood et al. 2016).

In the IPSL-CM5A model, we do not simulate markedly increased precipitation in the northern region of Greenland for this period. Thus, the initial surface mass balance at the beginning of PPT is still very weak.

Nonetheless, we agree that a figure showing the steps in the ice sheet growth over Greenland would be very informative. The figure has been added as Figure 3 in the paper.

Eiriksson, J., and Geirsdottir, A. 1991. A record of Pliocene and Pleistocene glaciations and climatic changes in the North Atlantic based on variations in volcanic and sedimentary facies in Iceland. *Marine Geology* 101: 147-159.

Geirsdottir, A., and Eiriksson, J. 1993. Growth of an Intermittent Ice Sheet in Iceland during the Late Pliocene and Early Pleistocene. *Quaternary Research* 42: 115-130.

Funder, S., Bennike, O., Mogensen, G.S., Noe-Nygaard, B., Schack Pedersen, S.A., and Petersen, K.S. 1984. The Kap København Formation, a late Cainozoic sedimentary sequence in North Greenland. *Rapp. Grønlands geol. Unders.* 120: 9-18.

Reviewer #2 (Remarks to the Author):

In their paper, Tan and coauthors present new model simulations of the evolution of the Greenland ice sheet during the Pliocene-Pleistocene transition (PPT) between 3.0 and 2.5 million years ago. Several proxies indicate that the initiation and intensification of the Northern Hemisphere (NH) glaciation occurred over this time period. Consequently, this time interval has received a lot of attention in the past decades. Many studies suggest that the atmospheric CO₂ concentration plays a major role in the initiation of glaciation both for the NH and for Antarctica. Since the available CO₂ reconstructions for the PPT are very uncertain, models have been used to explore the critical CO₂ threshold for Greenland and more generally NH glaciation. In this paper, the authors present transient simulations with an ice sheet model applied to Greenland and driven by snapshot climate model simulations for different boundary conditions (atmospheric CO₂, orbital configuration and Greenland ice volume). The model is then driven by different CO₂ scenarios, which are based on available reconstructions or inverse modelling results. They then use the ice rafted debris from several marine cores around Greenland to constrain plausible CO₂ pathways.

Response: Thank you for this concise summary of our study.

General comments

Although the study by Tan et al. presents interesting results and contributes to progress our understanding of the PPT and in particular the glacial inception over Greenland, I have some concerns about the methods and the conclusions which are drawn in the paper:

- No evaluation of the model is shown. Showing that the model can reproduce the present surface mass balance of the Greenland ice sheet would add credibility to the presented results. In particular there are different regional climate modelling results for the present day surface mass balance that could be used for the evaluation (e.g. MAR, RACMO). Although a good model performance

for the present day is not a guarantee that the model will perform well under very different conditions, it is a minimal requirement.

Response:

We agree that the validation of our model for present-day conditions is important and should be included in the paper. This was not done initially because the IPSL-CM5A climate model has been widely used for future and paleoclimate simulations (Dufresne et al. 2013, Kageyama et al. 2012, Contoux et al. 2012), including simulations of past Greenland ice sheets (Contoux et al. 2015, Tan et al. 2017).

However, we now have added diagnostics of the model validation under modern conditions in the Methods section (from line 334 to line 346). As suggested, we compare the simulated surface mass balance (SMB) obtained from the CMIP5 historical IPSL-CM5A simulation and averaged over the period 1981-2005 to the SMB computed with the regional state-of-the-art polar model MAR (Fettweis et al. 2013) forced by reanalysis over the same interval. The Figure 2 of this response (also added in the revised supplementary materials) shows that the simulated SMB with our climate model IPSL-CM5A is consistent with the reconstructed SMB from MAR, except in some regions of the northwest and southeast Greenland.

These differences can be attributed: 1) to the coarser resolution of IPSL-CM5A compared to MAR, which thus limits the model's ability to accurately simulate precipitation in mountainous regions (a common problem in fully-coupled climate models, see e.g. Sun et al. 2006), such as South Greenland here ; 2) to biases in simulated temperatures and precipitations in the Greenland region in IPSL-CM5A (Dufresne et al. 2013) and/or MAR. Nevertheless, the simulated SMB using IPSL-CM5A is reasonably close to that of MAR.

Although a good model performance for the present day is not a guarantee that the model will perform well under very different conditions, it is a minimal requirement.

Response:

We agree that validation of the model performances for modern conditions is not sufficient to guarantee its performance under different conditions. We thus evaluated our methodology and the model behavior against the more constrained Eemian deglaciation (LIG), as detailed in the response to the third reviewer, who explicitly asked for this evaluation. We thus invite you to refer to this response for full details on this evaluation simulation. The results show that our method consistently reproduces the waxing and waning of the Greenland ice sheet between 150 to 115 kyrs across the Eemian interglacial (please refer to Figure 6 in this response letter). The validation of our model results against GrIS modern conditions and the quantification of the sensitivity of our method against the Eemian glaciation support the use of this method to investigate Plio-Pleistocene Greenland ice sheet dynamics.

Figure.2 Simulated surface mass balance (mmWE/yr) averaged for the last 25 years (1981-2005) with IPSL-CM5A-LR AOGCM model (left) and MAR regional model (right)

- No model uncertainties are considered. I understand that large uncertainties in the evolution of the Greenland ice sheet over the PPT are associated with the poorly constrained CO₂ concentrations, but possible uncertainties in the model should also be quantified. In particular, the matrix method used to derive the climate forcing applied to the ice sheet model is based on a limited number of climate model simulations and cannot capture all non-linearities in the system. What would be the impact of having more/less climate states (e.g. an additional intermediate orbital configuration or less CO₂ concentrations)?

Response:

1) Concerning the matrix size:

The idea here is to provide a matrix spanning the range of possible values for the main GrIS forcings (insolation/CO₂/GrIS configuration), which subsequently allows the computation of the evolution of GrIS over time with a tri-interpolation procedure. In this paper, we provide 56 AOGCM simulations, representing an enormous effort in terms of work and computation time (~2.35Mhours). Ideally, the number of insolation, CO₂ or ice sheet configurations should be increased, but a compromise is required between the number of simulations used in the method and the computational costs needed to perform these simulations (see a detailed discussion in section 1.3 in Pollard 2010). Adding for instance an intermediate orbital configuration would require 28 additional AOGCM experiments, corresponding to the 7 ice sheet configurations and 4 different CO₂ values.

The of having more/less climate states is very difficult to infer. The climate states obtained with the coupled model somewhat act as tie points in the ice sheet simulation, in the sense that when the values of the forcings during the ice sheet simulation are close to the exact values prescribed in the climate states, the simulated GrIS is more constrained because the influence of the interpolation

procedure is reduced. In other words, the greatest deviations from our results if we add or remove climate states are thus expected to occur when the forcings are the most different from the values imposed in the climate simulations because this is where the interpolation method will have the largest imprint on the results. In terms of CO₂, our coverage of the range of possible values is already relatively wide, therefore we do not expect the addition of another CO₂ concentration to change results significantly. Adding an intermediate orbital configuration might have a larger impact but which configuration to choose is not obvious because an equivalent NH summer insolation value can be obtained from different configurations across the 3-2.5 Ma interval. On the contrary, the two “extreme” orbital configurations that we use correspond to those giving the minimum and maximum NH summer insolation values across the 3-2.5 Ma interval.

We stress however that the above inferences are hypothetical. They could be tested for instance with a faster climate model (but with a lower spatial resolution) . In this paper, we thus restrict the number of simulations to 56, which is, in our opinion, sufficient to provide interesting insights into the dynamic of the GrIS across the PPT.

2) Concerning uncertainties:

We agree with the referee that, indeed, there are uncertainties in the parameterization of the model (e.g., the lapse rate) and in the coupling method between climate and ice sheet models (e.g., the radius of interpolation, see Ladant et al. 2014), which must be taken into account. Following your comment (and that of referee 3), we have conducted sensitivity tests for these two relatively unconstrained parameters of the model.

Figure 3 presents the test results of sensitivity to three different lapse rates (-5 K/km, -6 K/km, -7 K/km) with different constant pCO₂ scenarios. Note that the lapse rate value of 6 K/km is the standard model value and is thus used in every GrIS simulation presented in this manuscript with the exception of these sensitivity tests. We find that the 5 K/km lapse rate only marginally alters the ice sheet evolution regardless of the CO₂ concentration. Using 7 K/km has a greater impact close to the CO₂ threshold, (~320 ppmv, Fig. 3b) as the simulated GrIS volume between 2.9 and 2.7 Ma is reduced by a factor of 2. Yet, the GrIS volume obtained at 2.7 Ma and the large decrease of the GrIS at 2.6 Ma does not depend on the lapse rate value. Figure 4 shows results of sensitivity tests to the lapse rate driven by the mean, minimum and maximum CO₂ estimates of Martinez-Boti et al. (2015). In a similar way to the constant CO₂ scenarios, the 7 K/km experiment consistently shows lower GrIS volumes but, importantly, the evolution pattern of the GrIS volume does not change. The conclusions and implications of our manuscript therefore hold true.

Figure 5 presents the results of sensitivity tests to the radius of interpolation in the ‘matrix’ method. We perform simulations with three different values (60 km, 75 km, 90 km). Although results are very similar between the three tested values, the larger radius results in a less dynamic ice sheet after 2.7 Ma. Again, the evolution pattern of the GrIS volume does not change. Here, 60 km was chosen as the standard radius for all the simulations.

Figure 3. Simulated GrIS volume with constant pCO₂ scenarios: 280 ppmv (a), 320 ppmv (b) and 360 ppmv (c). For each simulation, three different temperature lapse rates are tested: 5 K/km (purple), 6 K/km (green) and 7 K/km (orange).

Figure 4. Simulated GrIS volume with the pCO₂ scenarios of Martinez et al. (2015): mean values (a), minimum values (b) and maximum values (c). For each pCO₂ scenarios, three different temperature lapse rates are tested: 5 K/km (purple), 6 K/km (green) and 7 K/km (orange).

Figure 5. The simulated GrIS volume with the pCO₂ scenario of Martinez et al.(2015) mean values and different values for the radius of the interpolation (60km, 75km and 90km).

- In order to constrain the CO₂ evolution during the PPT, in the paper the authors compare the modelled Greenland ice volume with IRDs at different locations around Greenland. This comparison is very qualitative (Fig. 4b). Such a rough comparison can be used to rule out some extreme CO₂ scenarios, but nothing more. One way to improve the comparison would be to compare IRDs with calving from the ice sheet model into the ocean (if available). As also briefly discussed by the authors, the relation between IRDs and total ice volume is probably not linear. If the main aim of the paper, as mentioned in the title, is to constrain CO₂ pathways over the PPT, a figure showing all CO₂ reconstructions and their constrained range would be helpful.

Response:

We agree that the comparison between the modeled Greenland ice volume and IRDs is qualitative. This comparison can only be used to rule out some of the CO₂ scenarios. Therefore, we conclude from the comparison with IRD records that 3 of the scenarios (Martinez-Boti mean and high estimates and Seki high estimates) fit better than others with IRD records. We then suggest that the GrIS evolution driven by the Martinez-Boti mean CO₂ estimate is the best-case scenario based on observational evidence for ice sheets outside Greenland after 2.7 Ma (see Section 2.4 of the revised manuscript). To meaningfully improve the comparison with IRD records, we would need to add more processes at higher spatial resolution (such as for instance an iceberg model, see e.g., Smith et al. 2018), but also to extend the ice sheet domain to Northern Europe and America. Indeed, other ice sheets may play an important role after GrIS inception, for instance by contributing to IRD materials around Greenland, as shown by mineralogical studies (Jansen et al., 2000, and other reference). We believe that these questions would be exciting targets for future studies. Moreover, following a suggestion from the third reviewer, we have changed the title of the manuscript to “Sensitivity of the Greenland ice sheet to CO₂ during

the Plio-Pleistocene transition” to underline that our study explores the GrIS sensitivity to CO₂ across the PPT rather than really constraining CO₂ pathways.

- There is compelling evidence that after 2.7 Ma ice sheets were not limited to Greenland but covered also parts of Northern North America and Scandinavia during glacial times. The effect of ice covered areas outside of Greenland is not included in the model setup. The authors should at least discuss what implications that might have for the presented results.

Response:

This is a fair comment. In our study, the combination of high computational costs and complexity of implementation of the ‘matrix’ method does not allow us to simulate long-term ice sheet evolution over the entire Northern Hemisphere. However, we agree that after 2.7Ma, the onset of other regional ice sheets are likely to play an important role in the GrIS evolution. Through their radiative cooling impact on large regions of the Northern Hemisphere at least, the presence of ice sheets outside Greenland may help to maintain a large GrIS by counterbalancing the effects of increasing summer insolation and/or increasing pCO₂. In particular, in our GrIS scenarios, it may help limit the simulated GrIS deglaciation at 2.6 Ma. The revised manuscript provides a more thorough discussion on this question (from line 215 to line 226).

Please also note that, in the Eemian sensitivity simulation, we took into account LGM conditions that include the Laurentide and Fennoscandian ice sheets. This is warranted because the ice sheets of the last interglacial are better constrained than those of the late Pliocene.

Specific comments

Lines 148-153: reference to Fig. 3 is missing.

Response: Corrected.

Line 198: reference to Fig.3a-c seems inappropriate.

Response: Corrected.

Figure 2 is very hard to read. I would suggest splitting each panel into two, one showing the forcing and the other the response.

Response: We have considered this suggestion but we believe that it is important to have the orbital forcing and the GrIS evolution on the same panel to clearly see the combined evolution of both.

Figure 4b: why are there multiple points at each time for the IRD records?
English should be improved throughout the paper.

Response:

These IRD data come partly from the Pangea website and partly from the digitization of the figure. The multiple points do not exactly represent data points, they merely represent the trend in the data and the time axis for these multiple points is very close.

Reviewer #3 (Remarks to the Author):

This paper explores changes in the Greenland ice sheet through the Pliocene-Pleistocene transition. A coupled climate – ice sheet model is used to simulate the response of the ice sheet to varying orbital and CO₂ forcing. Several different CO₂ scenarios are used, corresponding to some idealised constant forcings and some scenarios from the published literature. Finally, the model-predicted ice volumes are compared with IRD records. The main conclusions that I took away are that (1) between certain CO₂ thresholds the ice sheet is very sensitive to orbital forcing, and (2) the Martinez-Boti and Seki CO₂ reconstructions give ice sheets that are most consistent with the IRD data.

Response:

These are indeed the major results of our study. The climate/ice sheet coupling allows us to test several pCO₂ reconstructions and to compare the resulting GrIS evolution with IRD records. Even if this comparison is qualitative, the model results for the evolution of the GrIS when forced by Martinez-Boti and Seki reconstructions are indeed in good agreement with the data.

Major Comments

In order to have confidence in the modelling strategy, an evaluation in the form of a simulation of a reasonably well known case study is needed, such as the last deglaciation (e.g. does a transient simulation of the last deglaciation end up with a reasonable modern ice sheet at present), or the last interglacial (e.g. how does the model compare with the Kopp Greenland record). In particular I am slightly concerned that only two different orbits (“warm orbit” and “cold orbit”) are used in the GCM that is used to construct the forcing matrix. This may result in a major simplification of the complexities of the orbital forcing over the time period 3 – 2.5 Ma. As such, some evaluation of the method is needed. Either this should be included in the paper, or it should be noted clearly in the paper that the methodology is relatively untested for Greenland.

Responses:

This is an important comment. Our method has been successfully used to capture the onset of the Antarctic ice sheet at the Eocene-Oligocene Transition (34 Ma, Ladant et al. 2014) but we agree that a sensitivity study in a more recent and constrained framework is important to assess our methodology. This is difficult to do for the last deglaciation, which is intricate, but it is possible to test our method on the Last Interglacial (LIG).

We have thus performed a sensitivity study for the Eemian deglaciation across the 150-110 Ka interval. To this end, we adapted our matrix of climatic states, by adding new experiments with lower CO₂ and larger GrIS to our late Pliocene matrix. We performed 4 additional climate simulations using an LGM ice sheet configuration, either 185 or 220 ppmv of CO₂, and our two extreme orbital configurations. We still restrict the ice sheet simulation to Greenland only and quantify the GrIS evolution and contributions to sea level during the LIG. Because the new simulations added in the matrix show extensive ice sheet and sea ice coverage over the Northern Hemisphere, we allow the ice sheet model to use variable basal melting factors in line with the multi-proxy index of Quiquet et al. (2013). Basal melting factors thus vary linearly between 0 and 5 m/year in regions with a depth of below 1300 m and are fixed to 10 m/year where depth is above 1300 m. In addition, we use the sea-level curve of Waelbroeck et al. (2002) as a forcing to determine which ice model grid points are floating and which are grounded. These two processes are added to take into account the climatic influence of the LIG deglaciation outside Greenland because we do not explicitly model this ice sheet evolution. Finally, the ice sheet model is initialized with the LGM GrIS.

The results from this sensitivity simulation are shown on Figure 6 of this response. As expected, the combined increase in insolation and pCO₂ leads to the melting of the GrIS between 133 and 123 Ka down to a minimal value of 2.846E15 km³. The timing of the deglaciation is consistent with results from other studies (e.g., Stone et al. 2013, Bradley et al. 2018, Goelzer et al. 2016). Stone et al. 2013 “best score” simulation places the start of the GrIS deglaciation at 135 Ka (their Fig. 6b, black lines) and the end around 123 Ka. Goelzer et al. 2016 restrict their figures to the interval 130-115 Ka so that the start of the deglaciation in their simulations is unclear but the lowest GrIS volume is also found around 123 Ka. Finally, Bradley et al. 2018 also place the beginning and end of the GrIS deglaciation at 133 and 123 Ka, respectively (their Fig. 2). The contribution of the GrIS to sea level rise during the LIG is poorly constrained and is estimated at between 0.6 and 3.5 m (Bradley et al. 2018). Our simulated minimal GrIS reaches a volume of 2.846E15 km³, which is slightly less than the equilibrium GrIS volume (2.87E15 km³) simulated by GRISLI under pre-industrial IPSLCM5A forcings. Compared to other studies, the volume of our simulated LIG GrIS is not sufficiently low. However, considering that the method is designed for the PPT and not specifically for the Eemian, we argue that the model does a reasonable job in reproducing the Eemian deglaciation. In summary, the reasonable agreement between our modeled deglaciation and the results from other transient LIG simulations instills confidence in our modelling strategy. We have included this evaluation simulation in the Methods section (line 347 to line 372).

Figure 6. The simulated GrIS volume based on the pCO₂ records from 150Ka to 110 Ka (green line; Petit et al.,1999). The blue line represents the summer insolation at 65N° (Laskar et al.,2004), the green line represents the Pco₂ records, the orange line is the simulated GrIS in this study.

It is well known that ice sheet models are highly sensitive to poorly-constrained model parameters. For example, Stone et al (2010) show very different ice sheet responses to CO₂ forcing depending on model parameters, all of which give good agreement with modern ice sheets under modern forcing. As such, the CO₂ thresholds quoted here are themselves somewhat poorly constrained. Gasson et al (2014) found a huge range in CO₂ threshold for Antarctic glaciation from different ice sheet models with different parameters. As such, I think it is very important either to explore this sensitivity in the paper, or it should be noted clearly in the paper that the results (and in particular the numerical values of the thresholds) may be highly dependent on the internal ice sheet model parameters.

Response:

We agree that tests on different parameters would provide firmer conclusions. However, we think that a sensitivity study of this scope merits a paper by itself, similar to the work of Stone et al. 2010.

In the present study, model parameters are identical to those used in the late Pliocene GrIS simulation of Contoux et al. 2015, with the exception of the resolution. We agree with the reviewer that the exact threshold values may be model dependent and have clarified this point in the conclusion of the revised manuscript.

Nevertheless, we now provide a few sensitivity tests on the topographic lapse rate and the radius of interpolation along the ice dimension of the matrix (see Ladant et al. 2014 for a full description of this method). These tests and the results are described in the response to reviewer 2, who has also raised this particular point (see Figures 3 to 5 of this response).

I think it would be very interesting to see a few longer simulations, that started earlier and finished later (e.g. from 3.5 Ma to 2 Ma). This would indicate whether the transition seen in these simulations is unique, or occurs at other time periods

with similar orbital forcing. Some of the CO₂ records used would be suitable for this.

Response:

In our initial study, we focused on the period 3-2.5 Ma. However, there are previous periods where GrIS grew and decayed, for instance MIS-M2 (Tan et al 2017), but this short period (50 ka) is poorly constrained in terms of pCO₂ changes. Nevertheless, it is interesting to enlarge our initial study period to one for which pCO₂ reconstructions are available. We have thus conducted longer simulations using the available pCO₂ data. First, we performed a simulation from 3.3 Ma to 2.3 Ma with the Martinez-Boti et al. (2015) mean CO₂ values (Figure 7). Second, we performed a simulation from 4.5 Ma to 2.5 Ma with Seki et al. (2010) low CO₂ estimates (Figure 8). Both simulations suggest that ice sheets have grown over Greenland prior to 2.7 Ma, although the most important Greenland glaciation happened around 2.7 Ma due to lower pCO₂ levels and insolation. This is consistent with studies suggesting that ice has been present, at least periodically, over Greenland for the latest part of the Neogene (e.g. Bierman et al. 2016). Our results also support findings from other studies (e.g. Schaefer et al. 2016), which suggest that Greenland may have experienced numerous intervals of deglaciation from 2.6 Ma onwards rather than being continuously glaciated.

Figure 7. Simulated GrIS from 3.4 Ma to 2.2 Ma with pCO₂ scenario of Martinez et al. (2015) mean values.

Figure 8. Simulated GrIS from 4.2 Ma to 2.5 Ma with the pCO₂ scenario of Seki et al. (2010) alkenone low data.

There were a lot of typos and grammatical errors. I noted these in a paper version, but the lack of line numbering in the submitted manuscript makes highlighting them all in this review very time-consuming. It also makes reviewing the manuscript harder, so please use line and page numbers in future.

Response:

We apologise for not using the line and page numbers. This was a regrettable oversight that has been corrected in our revised version. A concerted effort has been made to correct typos and grammatical errors.

Specific Comments

The title is unclear. How about “Sensitivity of the Greenland ice sheet to CO₂ at the Plio-Pleistocene transition”

Response: Thank you for this suggestion which we will adopt in the revised version.

The term “pathways” is odd...maybe “scenarios” or “records” when talking about the proxies.

Response: The term “pathways” has been replaced by “scenarios”.

Throughout, “favorable” should be clarified as “favourable for glacial inception”
Throughout “confronting” should be “comparing”

Response: Corrected, Thank you.

In the last line of the abstract, and towards the end of the conclusions, the relevance for future climate is briefly mentioned. This should be removed as it is not explored at all in the paper. Or alternatively it should be greatly expanded on in the main paper.

Response: We agree and have modified the sentences relating to future climates in the abstract and conclusion.

What is really meant by a “perennial” ice sheet should be clearly defined the first time it is used.

Response: We use the term “perennial” to mean “lasting several orbital cycles”. We have clarified it in the revised version (line 18).

I don’t understand the sentence that begins “Pioneering studies were carried out....”

Response: We have rewritten these sentences in the revised version (From line 60)

Summer insolation is not “perfectly” constrained, but “very well” constrained.

Response: We agree and replaced “perfectly” with “very well”.

Careful when discussing thresholds. E.g. “levels higher than 320” is stated when really all you know is that the threshold is somewhere between 320 and 360, it may be 359 ppmv!

Response:

We agree that the threshold value lies within the range of two consecutive pCO₂ concentrations modeled and cannot be narrowed down more than this. However, in this specific case, we think our formulation is justified because, according to our simulations, 320 ppmv is already higher than the maximum CO₂ value required to avoid a full deglaciation of the GrIS after 2.7 Ma. Therefore, at levels higher than 320 ppm, our simulations indicate that the GrIS cannot be maintained after 2.7 Ma.

The initial condition testing is done for the two extremes, but the most interesting cases are the intermediate CO₂ levels – it would be more informative to see the results from e.g. 320 ppmv with initial states of full and no ice sheet.

Response:

We agree. We have included the results from 320 ppmv with initial states of full and no ice sheet and have removed the extreme CO₂ cases. For more details please refer to the paper (Figure. 1b in the paper and starting from 109)

It is odd that you discuss the third record (Seki) first – maybe change the panel ordering of Figure 2.

Response: Corrected.

I don't agree that the comparison with Willeit et al “provides confidence in the ability of the model”. All it shows is that you get similar results to another modelling study with a similar model.

Response:

This sentence has been rewritten. (line 155)

However, although both studies are on modeling work, the methodologies used are totally different and climate models used are different (the CLIMBER EMIC forcing the regional model REMBO vs the fully coupled IPSLCM5A model).

The final paragraph before the conclusions is odd. I don't understand the sentence beginning “However, the integration” or the sentence beginning “Indeed, evidences”.

Response: We have reorganized the last part before the conclusions. Please refer to the paper for details (line 215 to line 226)

In the conclusion I don't understand why you can infer from your results that CO₂ has to be in a narrow window, or really what "trigger and maintain" really means.

Response:

We meant that after 2.7 Ma, the CO₂ concentrations have to remain within a narrow range of values (< 320 ppm) for the GrIS to develop and remain perennial (i.e., not fully melt). We have rewritten the conclusions to clarify several points.

Figure 4: For the IRD, it is not clear if an absence of data (e.g. around 2.55 Ma to 2.6 Ma at site 611) means no IRD, or just no record.

Response: According to the observational reference, the absence of data means no IRD records.

Figure S3 caption – presents should be represents.

Response: Done.

Figure 2 – what are the uncertainties in CO₂? Do they represent a 1-sigma, 2-sigma, or range, or what?

Response:

In Martinez et al., 2015, the uncertainty range encompasses 68% of 10,000 Monte Carlo simulations performed to fully propagate uncertainties. In Bartoli et al. (2011), the uncertainty range is 1-sigma from the average. In Seki et al. (2010), the two alkenone-based CO₂ reconstructions are obtained using size-corrected $\epsilon_{37:2}$ values for the modern range of b values (b is a term that summarizes physiological factors such as growth rate or cell geometry). This information can be found in the legend of Figure 2 in the paper.

Stone et al (2010). Investigating the sensitivity of numerical model simulations of the modern state of the Greenland ice-sheet and its future response to climate change. *The Cryosphere*, 4, 397-417.

Gasson et al (2014). Uncertainties in the modelled CO₂ threshold for Antarctic glaciation, *Clim. Past*, 10, 451-466.

References

Bierman et al. (2016). A persistent and dynamic East Greenland Ice Sheet over the past 7.5 million years. *Nature*, 540(7632), 256.

Bradley et al. (2018). Quantification of the Greenland ice sheet contribution to Last Interglacial sea level rise. *Clim. Past* 9, 621–639 (2013).

Contoux et al. (2012). Modelling the mid-Pliocene Warm Period climate with the IPSL coupled model and its atmospheric component LMDZ5A. *Geoscientific Model Development*, 5(3), 903-917.

Contoux et al. (2015). Modelling Greenland ice sheet inception and sustainability during the Late Pliocene. *Earth and Planetary Science Letters*, 424, 295-305.

Dolan et al. (2012). Pliocene ice sheet modelling intercomparison project (PLISMIP)–experimental design. *Geoscientific Model Development*, 5(4), 963-974.

Dowsett et al. (2013). Sea surface temperature of the mid-Piacenzian ocean: a data-model comparison. *Scientific Reports*, 3.

Dufresne et al. (2013). Climate change projections using the IPSL-CM5 Earth System Model: from CMIP3 to CMIP5. *Climate Dynamics*, 40(9-10), 2123-2165.

Fettweis et al. (2013). Estimating the Greenland ice sheet surface mass balance contribution to future sea level rise using the regional atmospheric climate model MAR. *The Cryosphere*, 7, 469-489.

Goelzer et al. (2016). Impact of ice sheet meltwater fluxes on the climate evolution at the onset of the Last Interglacial. *Clim. Past* 12, 1721–1737 .

Haywood et al. (2016). The Pliocene Model Intercomparison Project (PlioMIP) phase 2: scientific objectives and experimental design. *Climate of the Past*, 12(3), 663-675.

Kageyama et al. (2013). Mid-Holocene and Last Glacial Maximum climate simulations with the IPSL model—Part I: Comparing IPSL_CM5A to IPSL_CM4. *Climate dynamics*, 40(9-10), 2447-2468.

Ladant et al. (2014). The respective role of atmospheric carbon dioxide and orbital parameters on ice sheet evolution at the Eocene-Oligocene transition. *Paleoceanography* 29, 810–823.

Laskar et al. (2004). A long-term numerical solution for the insolation quantities of the Earth. *Astron. Astrophys.* 428, 261–285.

Martinez-Boti et al. (2015). Plio-Pleistocene climate sensitivity evaluated using high-resolution CO₂ records. *Nature* 518, 49–54. doi:10.1038/nature14145

Morlighem et al. (2017). BedMachine v3: Complete bed topography and ocean bathymetry mapping of Greenland from multibeam echo sounding combined with mass conservation." *Geophysical research letters* 44.21 (2017).

Petit et al. (1999). Climate and atmospheric history of the past 420,000 years from the Vostok ice core, Antarctica. *Nature*, 399, 429–413.

Pollard et al. (2010). Modelling West Antarctic ice sheet growth and collapse through the past five million years. *Nature*, 458(7236), 329–332.

Quiquet et al. (2013). Greenland ice sheet contribution to sea level rise during the last interglacial period: a modelling study driven and constrained by ice core data. *Climate of the Past*, 9(1), 353-366.

Ritz et al. (2001). Modeling the evolution of Antarctic ice sheet over the last 420,000 years: Implications for altitude changes in the Vostok region. *Journal of Geophysical Research: Atmospheres*, 106(D23), 31943-31964.

Schaefer et al. (2016). Greenland was nearly ice-free for extended periods during the Pleistocene. *Nature*, 540(7632), 252.

Stone et al. (2013). Greenland ice sheet contribution to Last Interglacial sea level rise. *Clim. Past* 9, 621–639.

Sun et al. (2006). How often does it rain? *Journal of Climate*, 19(6), 916-934.

Tan et al. (2017). Exploring the MIS M2 glaciation occurring during a warm and high atmospheric CO2 Pliocene background climate. *Earth and Planetary Science Letters*, 472, 266-276.

Taylor et al. (2012). Taylor, K.E., Stouffer, R.J. and Meehl, G.A., 2012. An overview of CMIP5 and the experiment design. *Bulletin of the American Meteorological Society*, 93(4), pp.485-498.

Waelbroeck et al. (2002). Sea-level and deep water temperature changes derived from benthic foraminifera isotopic records. *Quaternary Science Reviews*, 21(1-3), 295-305.

REVIEWERS' COMMENTS:

Reviewer #1 (Remarks to the Author):

I think you have answered the questions as far as possible. In any modeling study there are always a certain set of assumptions but the present revised paper is an important addition to the literature.

Reviewer #3 (Remarks to the Author):

First of all, thanks to the authors for their careful and thorough response to the original reviews. In my opinion, the paper is very close to being ready for publication, but I have some additional comments:

(1) The references need some work. E.g. lines 449, 472, 432, 401, 414, 387 have typos. Refs 22, 27, 39, 43 and 49 are Discussion papers and should be updated or removed.

(2) Line 24-27. I still don't like this sentence. How about "...evolution across the PPT, we highlight the pivotal role of pCO₂ on the GrIS expansion. In particular, our model results indicate that pCO₂ levels less than about 280 ppmv allow the development of a full GrIS, whereas pCO₂ higher than 320 ppmv prevent the GrIS from remaining perennial". **but see comment (8) and (9) below**

(3) line 54. Give dates of MIS-M2.

(4) "snapshots" should be "snapshot"

(5) line 66. "started from" should be "the underlying GCM simulations included a"

(6) Figure S1 – panels need labelling a, b, c etc.

(7) Figure 1 – caption needs to use same nomenclature as the figure legend, so explain what e.g. ISV_220 is.

(8) I still don't agree with the summary of the thresholds. All you can say from these simulations is that the threshold lies somewhere between 280 ppmv and 240 ppmv (not "under 280 ppmv"). Also, the maximum threshold is somewhere between 320 and 360 (not "above 320").

(9) Line 101. Similarly here, I would say that you have constrained the "narrow range" to between 240.0001 to 359.999 ppmv, so more like 120 ppmv than 50 ppmv !! this should also be clarified in the abstract.

(10) Figure 1b. Those simulations which are identical to those in Figure 1a should be in the same colour as in Figure 1a.

(11) Figure 2b. At ~2.6 Ma, the mean red line is actually below both the maximum and minimum red lines, so the red shading disappears. Need to add the red shading around here.

(12) Line 154. Reference for IRD records.

(13) Line 156. Remove "best fit".

(14) Line 168. "the primary location" should be "one of the primary locations" (e.g. Baffin Island).

(15) Line 344, Figure S4 – please show a difference plot as well as the two panels, so we can be

sure where the main differences really are.

(16) Line 364. A plot fo the minimum Eemian ice sheet extent and height would be very informative.

(17) Line 367,370. Remove scientific notation "E".

Responses to reviewers' comments for

“Dynamic Greenland ice sheet driven by pCO₂ variations across the Pliocene Pleistocene transition”

Reviewer #1 (Remarks to the Author):

I think you have answered the questions as far as possible. In any modeling study there are always a certain set of assumptions but the present revised paper is an important addition to the literature.

Response: Thank you for this comment.

Reviewer #3 (Remarks to the Author):

First of all, thanks to the authors for their careful and thorough response to the original reviews.

Response: Thank you for your comments which largely help to improve our manuscript.

In my opinion, the paper is very close to being ready for publication, but I have some additional comments:

(1) The references need some work. E.g. lines 449, 472, 432, 401, 414, 387 have typos. Refs 22, 27, 39, 43 and 49 are Discussion papers and should be updated or removed.

Response: Thank you for these detailed reviews. We have corrected the typos in the references and updated all discussion papers.

(2) Line 24-27. I still don't like this sentence. How about "...evolution across the PPT, we highlight the pivotal role of pCO₂ on the GrIS expansion. In particular, our model results indicate that pCO₂ levels less than about 280 ppmv allow the development of a full GrIS, whereas pCO₂ higher than 320 ppmv prevent the GrIS from remaining perennial". **but see comment (8) and (9) below**

Response: Thank you for this comment. According to the editorial policy, we must limit the abstract within 165 words, so the previous sentence beginning with "in particular" has been deleted. We address this comment in the first part of the Results. Please refer to the answer to comment (8).

(3) line 54. Give dates of MIS-M2.

Response: Done.

(4) "snapshots" should be "snapshot"

Response: Corrected.

(5) line 66. “started from” should be “the underlying GCM simulations included a”

Response: Corrected.

(6) Figure S1 – panels need labelling a, b, c etc.

(7) Figure 1 – caption needs to use same nomenclature as the figure legend, so explain what e.g. ISV_220 is.

Response: Done, we added more explanations for the labels.

(8) I still don't agree with the summary of the thresholds. All you can say from these simulations is that the threshold lies somewhere between 280 ppmv and 240 ppmv (not “under 280 ppmv”). Also, the maximum threshold is somewhere between 320 and 360 (not “above 320”).

Response: We agree that our experiments do not unambiguously ascribe precise CO₂ levels to the minimum and maximum thresholds. In our study, the thresholds for the onset and maintenance of a large perennial Greenland glaciation ice sheet are indeed comprised between 240 and 280 ppmv and between 320 and 360 ppmv for the minimum and maximum CO₂ thresholds respectively. We have rephrased the sentence as: “..., the evolution of the GrIS clearly shows that below 280 ppmv of pCO₂, it is possible to trigger and maintain a large perennial ice sheet over Greenland even during intervals of strong summer insolation, in particular around 2.6 Ma, whereas pCO₂ higher than 320 ppmv prevent the GrIS from remaining perennial across the whole PPT period.”

We think that this wording better illustrate our meaning. Indeed, we agree that the use of the word “threshold” was not fully appropriate. Here, we simply state that at 280 ppmv, a perennial GrIS is simulated, as ice remains over Greenland even at 2.6 Ma (~ 8.10¹⁴ km³ in volume). Therefore, below this value, our model simulates perennial GrIS. On the contrary, the GrIS almost disappears (retreating to ~ 2.10¹⁴ km³) at 2.6 Ma under 320 ppmv of CO₂. At CO₂ > 320 ppmv, the GrIS is thus not perennial across the whole PPT period (3 – 2.5 Ma). We also modified some details (e.g. replace “under” with “below”, change the pCO₂ range from “between 280 and 320 ppmv” to “between 240 and 360 ppmv”) and deleted the narrow windows of 50ppmv in this paragraph. We then change the subheading from “Thresholds” to “GrIS sensitivity to the constant pCO₂ scenarios.”

(9) Line 101. Similarly here, I would say that you have constrained the “narrow range” to between 240.0001 to 359.999 ppmv, so more like 120 ppmv than 50 ppmv !! this should also be clarified in the abstract.

Response: Please refer to the answer to the previous comment.

(10) Figure 1b. Those simulations which are identical to those in Figure 1a should be in the same colour as in Figure 1a.

Response: Corrected.

(11) Figure 2b. At ~2.6 Ma, the mean red line is actually below both the maximum and minimum red lines, so the red shading disappears. Need to add the red shading around here.

Response: Done.

(12) Line 154. Reference for IRD records.

Response: Done.

(13) Line 156. Remove “best fit”.

Response: Corrected.

(14) Line 168. “the primary location” should be “one of the primary locations” (e.g. Baffin Island).

Response: Corrected.

(15) Line 344, Figure S4 – please show a difference plot as well as the two panels, so we can be sure where the main differences really are.

Response: We have added a difference plot in Supplementary Fig.4.

(16) Line 364. A plot for the minimum Eemian ice sheet extent and height would be very informative.

Response: We have added this figure in Supplementary Fig.6.

(17) Line 367,370. Remove scientific notation “E”.

Response: Corrected.